# Development of an intervention to support reproductive health of garment factory workers in Cambodia: a qualitative study

Chris Smith ,[1,2] Ly Sokhey,[3] Camille Florence Eder Tijamo ,[3] Megan McLaren,[4] Caroline Free,[5] Justin Watkins,[6] Ou Amra,[3] Chisato Masuda,[1] Elisa Oreglia[7]

For numbered affiliations see end of article.

**Correspondence to**
Dr Chris Smith;
christopher.smith@lshtm.ac.uk

## ABSTRACT

**Objectives** The aim of this study was to describe the development of an intervention to support the reproductive health of garment factory workers in Cambodia.

**Design** A qualitative study informed by intervention mapping which included semistructured interviews and participant observation, followed by intervention development activities including specifying possible behaviour change, designing the intervention, and producing and refining intervention content.

**Setting** The research was conducted by a multidisciplinary team with backgrounds in public health, linguistics, digital cultures and service delivery in a suburb of Phnom Penh where many garment factories cluster.

**Participants** Garment factory workers in Cambodia; typically young women aged under 30 years who have migrated from rural areas to Phnom Penh city.

**Outcomes** Analysis of reproductive health issues facing garment factory workers and metrics of videos developed.

**Results** Our research identified some challenges that Cambodian garment factory workers experience regarding contraception and abortion. Concerns or experience of side-effects were identified as an important determinant leading to non-use of effective contraception and subsequent unintended pregnancy. Financial constraints and a desire to space pregnancies were the main reported reasons to seek an abortion. Information about medical abortion given to women by private providers was often verbal, with packaging and the drug information leaflet withheld. In order to address limitations in the provision of accessible reproductive health information for factory workers, and given their observed widespread use of social media, we decided to make three 'edutainment' videos about family planning. Key social media metrics of the videos were evaluated after 1 month.

**Conclusions** We describe the development of an intervention to support reproductive health among garment factory workers in Cambodia. These videos could be further improved and additional videos could be developed. More work is required to develop appropriate and effective interventions to support reproductive health of garment factory workers in Cambodia.

## Strengths and limitations of this study

► This study sought to understand the reproductive health issues of a potentially vulnerable population of garment factory workers in Cambodia.
► This study was conducted by a multidisciplinary team with backgrounds in public health, linguistics, digital cultures and service delivery.
► We sought to employ participatory approaches in our needs assessment and intervention development.
► Being a qualitative study, the number of participants was relatively small and may not be representative of garment factory workers or health providers.
► It was challenging to explain and translate details of intervention mapping across languages, cultures and disciplines working in different paradigms.

## BACKGROUND

Cambodia has made considerable improvements in economic and health indicators in the last two decades, with much of Cambodia's economic growth attributed to garment exports.[1 2] In 2015, the estimated number of garment factory workers in Cambodia was estimated to be around 650 000, of whom 85% were female.[3 4] Garment factory workers are potentially a vulnerable population in terms of support for reproductive health issues and access to services. A study of female factory workers conducted in 2014 reported that 80% were younger than 30 years, half had only primary school education, and most had migrated from rural areas away from their family and community support and did not have adequate access to affordable, high-quality reproductive health information and services to meet their diverse needs.[5]

In Cambodia, reproductive health services are available through the public and private sectors, including non-governmental organisations (NGOs) and promoted through

traditional and social media, and through community initiatives such as Marie Stopes Ladies.[6–8] A wide variety of contraception methods are available. Abortion was legalised in 1997 and medical abortion was approved by the Cambodian Ministry of Health in 2009.[9] Previous studies have examined medical abortion provision in clinics and pharmacies. A study in 2015 reported good knowledge of medical abortion among healthcare providers in clinics and pharmacies in Cambodia.[9] A study of clinical outcomes of medical abortion in 2019 reported no increased need for additional treatment among women seeking medical abortion from pharmacies compared with clinics, but that women were less likely to feel prepared for what happened after taking the pills or to be offered contraceptive methods when getting medical abortion pills. Users of the different services varied with 26% of women in the pharmacy group being garment factory workers compared with 9% in the clinic group.[10]

This current study was conceived based on the prior observation that many independent private healthcare providers are located in the vicinity of garment factories, but not much is known about what kind of reproductive health services they provide, particularly medical abortion and contraception, and the experience of clients who visit them. The project is a collaboration between Marie Stopes International Cambodia (MSIC) and three universities in the UK. The study protocol and description of family planning practices are described elsewhere.[11 12] The aim of this paper is to describe the development of an intervention to support the reproductive health of garment factory workers in Cambodia.

## METHODS

This qualitative study was conducted during January 2018–December 2019 in a suburb of Phnom Penh where many garment factories cluster, with the cooperation of three factories and a number of private healthcare providers. We were a multidisciplinary and multinational team (Cambodian and UK-based researchers) with backgrounds in public health, linguistics, digital cultures (that is, the study of how different people and groups engage with digital technologies) and service delivery. The research was conducted by different members of the research team at different times, as illustrated in figure 1. Each member focused on their area of expertise and methods, while participating in data gathering done by other members. Findings from the different phases and approaches informed subsequent data gathering, in an iterative process aimed at understanding the wider context of medical abortion and contraception first (needs assessment), and develop possible interventions later. Research methods included general and participant observation, semistructured interviews and periodical meetings in the UK and Cambodia to discuss findings.[11]

The methods were informed by intervention mapping methods,[13] but in a more holistic manner, which took into consideration the different disciplines involved. We aimed to conduct a needs assessment followed by intervention development activities including specifying possible behaviour change targets, designing the intervention, and producing and refining intervention content. We transcribed and translated all interviews, and coded them both deductively, using codes derived from the structure of the intervention mapping needs assessment process, and inductively, using codes that emerged from the more open-ended methods, including observations and semistructured interviews with factory workers. At least two different researchers coded each interview, one deductively and one inductively. The end result was not a pure application of intervention mapping but rather a modified approach to it.

**London and Phnom Penh Fieldwork | Timeline 2018-2020**

| | | JAN | FEB | MAR | APR | MAY | JUN | JUL | AUG | SEP | OCT | NOV | DEC |
|---|---|---|---|---|---|---|---|---|---|---|---|---|---|
| **2018** | | | London: development of research protocols | | Phnom Penh: observation at factories, qualitative interviews at healthcare providers, analysis of MSI social media strategy | | | | Phnom Penh: interviews with health providers and women seeking MA | | | | |
| | | | Ethics protocols in London and Phnom Penh approved | | | | | | Phnom Penh: participant observation and "hanging out" at factories. Qualitative interviews with factory workers, mobile phone sellers, mobile services providers, factory infirmaries | | London & Phnom Penh: interview transcription and first round of inductive and deductive coding; revision of qualitative interview protocols. | | |
| **2019** | | Phnom Penh: all reserchers discuss findings, theory, gaps for remaining fieldwork | Phnom Penh: development of videos informed by research; iteration and testing among factory workers. | | | | London: Youtube keyword search 1 | | London: Youtube keyword search 2 | | London: data analysis, paper drafts | Phnom Penh: last observations in factories and public spaces, MSI social media & call center analysis, MSI instructional video development | London: data analysis and paper writing |
| | | Phnom Penh: follow-up interviews with providers and factory workers | | Phnom Penh: interviews with health providers | | | | | | | | | Phnom Penh: Youtube keyword search 3 |

**Figure 1** London and Phnom Penh fieldwork timeline. MA, medical abortion; MSI, Marie Stopes International.

We began with a short ethnographic study of female factory workers' lives, living conditions, work places and digital technology use, in order to learn about the wider context of choices regarding reproductive health. We selected three different factories with which our local partners had an existing relation, and obtained their permission to visit the factories during working hours to chat with workers. We introduced ourselves as researchers to the factory workers, and were not accompanied by factory representatives during such interactions. We joined several women, individually or in small groups, during their lunch breaks in factory grounds, chatting about their experiences and getting a sense of who they were friends with, what they discussed during breaks, how they found their current jobs and what their future plans were. We also shared meals with them in nearby restaurants after working hours and at weekends, and were able to visit some of their homes. After an initial period of what ethnographers call 'hanging out', that is, unstructured time spent with research participants doing whatever they do, in order to become familiar with their lives and to give them time to get used to our presence,[14] we conducted 17 semistructured interviews, using snowball sampling, that focused primarily on their mobile phone use, to understand what sites/apps they used, what content they consumed online and off, and whether they searched for specific information. Some of the interviews covered healthcare and health-related topics. We also did three semistructured interviews with owners of mobile phone shops and service providers near the factories, to triangulate what we learnt from the interviews about mobile phone use.

At the same time, we observed the practice of and interviewed garment factory infirmary workers and a variety of private healthcare providers ranging from simple drug stores to larger clinics, in order to learn about garment factory workers' use of reproductive health services. In this phase of research, we conducted 16 interviews with women seeking abortion from private providers located near the factories, focusing on reasons for seeking abortion services and their experiences, and 13 interviews with private providers working in different facilities such as infirmaries, pharmacies and bigger clinics, to understand practices around reproductive health, especially with regard to contraception and abortion provision (online supplemental file 1). Findings from these interviews are reported elsewhere.[12]

During the course of the project, we organised several meetings to share and discuss behaviours observed in research that could feasibly be targeted and potential interventions resulting from the needs assessment activities, and then design, produce and refine a pilot intervention. In order to address limitations in the provision of accessible contraceptive information and information materials for factory workers in general and specifically at the time of seeking medical abortion, we decided to make three 'edutainment' videos about family planning and one instructional video about how to take medical abortion correctly. We collaborated with a Cambodian media company (Phare) and during the process sought input from garment factory workers in a pretesting workshop.[15] The videos were designed to target barriers to contraception use that were identified in the needs assessment, ethnographic and interviews studies.[12] The videos were released through MSIC's communication channels and their impact was evaluated by an independent digital media company (Havas Champagne media).[16]

## Patient and public involvement

Neither patients nor the public were involved in the development of the research question, study design or outcome measures. Study participants had some involvement in the conduct of the study as follows: private providers were involved in recruitment of women seeking abortion services, and factory workers were involved in the production of the intervention through contributing to the scripts, acting in the videos and providing feedback prior to their dissemination. We thank everyone who participated in this research project.

## RESULTS
### Research context

A summary of sources of information to gather information for the needs assessment is shown in table 1. Through observation, discussion and the limited available literature, we were able to learn about the context and lives of garment factory workers in Phnom Penh.[17] There were several factories in the area each employing several thousand workers, typically working 6 days per week. Workers sat at sewing machines or did other tasks at individual work stations. Between the noise and the factory floor lay-out, there were limited opportunities to talk among colleagues during working hours. Meal times were when workers gathered in small groups, with different people (different ages and different places of origins) coming together rather than a fixed group of friends. The breaks tended to be short, and people either chatted or looked at their phones while having their meals. In terms of living conditions, workers who came from far away areas rented rooms in apartment blocks near the factories, sharing with family or other coworkers, often from their same area. Apartments were carved up in different rooms typically rented separately, so space was very limited and privacy of any kind hard to come by. While nearby guest houses provided the opportunity for discreet romantic relationships, factory workers tended to be surrounded by other people at all times. Shops around factories catered to the needs of the mostly young, hard-working residents: food and daily necessities shops, mobile phone sellers and service providers, and licensed and unlicensed healthcare providers, from simple drug-stores to clinics with consultation rooms and beds. There were no government or NGO clinics in the immediate vicinity of our research area, although some private providers had previously received training from NGOs

**Table 1** Sources of information for the needs assessment

| Information domain | Published and grey literature | Observation | Interviews (including discussions) |
|---|---|---|---|
| Factory worker demographics | 70% are aged <30 years, half have only primary school education, and most of them have migrated from rural areas away from their family and community support. Higher rate of abortion compared with young women in the national survey, contraception use | Observations of factory workers in and outside of factories consistent with literature | Interviews consistent with literature and observations, although not randomly selected. |
| Factory workers' daily life and living conditions outside the factory | Grey literature (NGO reports and newspaper articles) but very limited in detail. Academic literature focused on specific aspects of health and earnings. Limited qualitative work highlighting living conditions (Chansanphors 2008) | Observations in factories (working space, canteen, infirmary, public spaces), in shops and markets near factories and in 1 worker's home. This provided a useful perspective in the spaces where workers live, what kind of goods and services they have access to, how far they need to go to access specific services. | Formal interviews with 33 female factory workers. 'Hanging out' at break times, over meals and at their homes for informal conversations. This allowed us to build rapport, and to contextualise, triangulate and clarify information received from a variety of sources. |
| Sources of information for family planning and abortion | Literature and grey literature on contraception and abortion needs and services. No research exists on use of web sources for health/sexual health information in Cambodia | ▶ Direct observation of online activities did not yield any finding regarding searching for family planning information.<br>▶ Direct search on YouTube for 'family planning', 'abortion', 'medical abortion', 'contraceptive pill' showed several videos on the topics, some instructional, some editorial. All videos had comments, many from 2019 (regardless of when the video was first published), which indicate increasing engagement with online sources to look for family planning information. | Interviews with factory workers indicated a strong reliance on family and friends for information related to contraception and abortion. Medical practitioners were also cited as a source of information, but less influential. Interviewees who were asked directly denied looking for family planning information online, but some said they looked for other health information on Facebook or YouTube. |
| Family planning providers' reproductive health practices | Published literature on family planning in Cambodia | Observation in garment factory infirmaries and private providers. | Interviews with 22 providers, including factory nurses, pharmacists, private nurses and doctors. |

NGO, non-governmental organisation.

through social marketing or social franchising activities. The factories we visited all had infirmaries for first-aid and minor ailments. Mobile phone shops sold new and used mobile phones, ranging from US$15 for an old style feature phone to US$200 for a smartphone. Specialised shops created Facebook accounts on behalf of their clients, did small repairs, downloaded apps and content such as film and music, cheaper than using contract data to download online content. To complete the general picture of the life of factory workers, it is worth noting that as of January 2019, the minimum monthly wage was US$182 per month,[18] which could increase with overtime or for skilled workers, or be lower if workers did fewer hours or worked for workshops instead of established

factories. Our interviewees reported paying an average rent of US$35 per month (for a shared room) plus about US$5 in utilities. Finally, garment factory workers tended to be more in debt than the average Cambodian (40% of factory worker households were reported to be in debt, vs a national rate of 37%).[19]

## Reproductive health

The interviews with women seeking abortion from private providers, and with the private providers showed that in brief, for health issues during worktime, workers were able to use the factory infirmary, and outside of work the option to use public or private providers. The main role of the infirmaries was to treat minor ailments. Only the largest factory provided contraception. Infirmary staff reported that women would seek consultation elsewhere in the event of an unintended pregnancy.

Factory workers' opportunities for using contraception to prevent pregnancy were constrained by their long working hours, limiting their ability to attend distant clinics with restricted opening hours, for example, for long-acting contraception. Pharmacies close to their home and workplace were therefore the most practical option for healthcare seeking but there was sensitivity and stigma related to obtaining contraception or medical abortion near where women lived and worked. Women seeking supplies from drug stores or pharmacies would try to go to places far from where they lived to avoid being recognised, or would need to wait until no one else was at the counter or whisper to the provider to achieve any privacy. Young, unmarried women sought privacy in both buying and taking contraception such as the pill because of negative social attitudes towards sexual activity for unmarried women, yet privacy was very difficult to achieve in cramped and shared living conditions, and in small neighbourhoods where everybody knows everybody. Young women either shared a room with other young women or sometimes lived with an older female relative. There was also the suggestion that young women in relationships with older, wealthier men (meeting in nearby guest houses) might be unwilling or unable to insist on condom use. For married women, long periods living away from partners meant that contraception was only needed intermittently and unpredictably.

Regarding contraception use in general, women reported receiving information from a variety of sources such as family, friends, healthcare providers or the media. Women had concerns about contraceptives. Some women reported discontinuing contraception due to experience of side-effects such as 'body heat' sensation, weight loss due to reduced appetite, vomiting, menstruation changes, skin changes, dizziness and fatigue. Some women had never used contraception due to fear of side-effects including fear of infertility.

Financial constraints and a desire to space pregnancies were the main reported reasons to seek an abortion. In most cases, women obtained information about abortion from family and friends. In a few cases, women went directly to a clinic. None reported seeking information via the internet. Although most private providers were owned by a medical professional such as a doctor or pharmacist, the day-to-day running was often done by a nurse or pharmacist who was a family member or hired staff. Most had undergone formal training but some had learnt on the job or through short training courses. There was limited use of treatment guidelines or protocols. Several providers expressed a wish for more training, including Comprehensive Abortion Care training. All of the providers offered short-acting contraceptives and medical abortion. Some larger clinics offered ultrasound, surgical abortion (vacuum aspiration) and had beds for patients. A variety of medical abortion products were available in the private providers around the factories. In some cases, someone would request the drugs on behalf of another person. Information given to women was often verbal, with packaging and the drug information leaflet withheld, sometimes upon request of the women to protect their privacy at home, but also to prevent clients from knowing the brand name and looking for the drug elsewhere. In the cases where a drug information leaflet was given, it was not always in the Khmer language nor had pictorial instructions. Hence, women often had to remember the sequence of taking the drugs, side-effects and potential warning signs. Information on post-abortion family planning was variable and follow-up was generally in the case of problems rather than routine.

## Videos about family planning in Cambodia

Given the observed widespread use of Facebook and YouTube among factory workers, and the preference for receiving information via images, video and voice rather than text that emerged from interviews and observations, we searched YouTube for information on contraception, abortion and medical abortion, using a variety of search terms that reflected both medical and more popular terms. There were few results, and none of the videos we found had been posted by NGOs or government sources, possibly because such videos, although they exist, are not optimised for Khmer language and thus not findable when searching in Khmer. The videos that existed were not professionally produced, often consisting of little more than a slide show or a static image with a voice-over, but they were very descriptive and factual, and were beginning to show views in the thousands and tens of thousands. A detailed analysis of this phase of the research will be reported elsewhere, but our working hypothesis, based on YouTube searches, interviews with factory workers and mobile providers, and the increasing number of mobile internet and social media users in the country, estimated to have reached 12 million over a population of 16.8 million in January 2021,[20] was that the use of the internet to seek health information is just at the beginning for the large number of factory workers who are just starting to go online, and that it will increase significantly in the next few years.

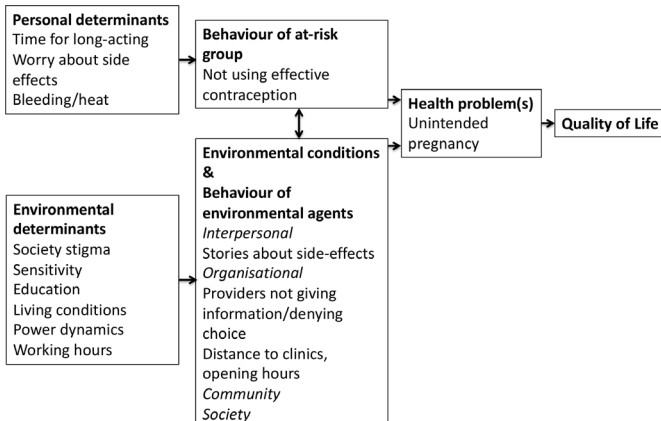

**Figure 2** Logic model of the problem.

## Intervention development

We held a midpoint workshop in Cambodia in order to discuss the research findings and their implications. Findings were also discussed in separate meetings with stakeholders in the Cambodian Ministry of Health and other NGOs. Consistent with previous research in Cambodia, we identified concerns or experience of side-effects as an important determinant leading to non-use of effective contraception and subsequent unintended pregnancy.[21 22] These experiences or concerns were shared by potential users of contraception and healthcare providers. Some of the reported side-effects such as 'body heat' or 'weight loss' (as opposed to weight gain) did not appear to be reported in standard contraception provider guidance or materials developed for potential users. We discussed the possibility that Cambodian women could experience more pronounced hormonal side-effects compared with western women due to differences in size and metabolism. This hormonal 'mismatch' theory has been described elsewhere,[23] but it was beyond the scope of our study to explore this systematically. We also discussed similarities and differences between Cambodian women's reports and those of women in the UK talking about hormonal contraception being 'unnatural' making them feel 'out of balance', which again is not widely raised in materials developed for potential users.[24] An attempt was made to construct a logic model of the problem, as per intervention mapping techniques (figure 2). In this model, we considered the health problem to be unintended pregnancy (clearly for women who did not want the pregnancy), and the 'at-risk' population to be factory workers with unmet need for contraception. We considered behaviours of the at-risk group and environmental agents, and determinants leading to those behaviours.

A summary of potential interventions is shown in table 2 using some terminology from the WHO classification of digital health interventions.[25] We felt interventions aimed primarily at providers would be challenging due to the number of different providers, their competing financial and business concerns and the need for sustained effort and follow-up, and was beyond the scope of the project. It was apparent that some providers had received training on reproductive health topics such as contraception and abortion from NGOs through social marketing or other programmes. Most providers were interested in more training. We facilitated additional training for those that requested it but it was beyond the scope of this project to provide a comprehensive training package. There was also interest from providers in peer-to-peer support for questions and advice leveraging social and informal ties,[26] such as through social media or instant messaging, potentially moderated by a reproductive health expert from an NGO or health department. There did not seem to be any capacity to do this at scale at the time of the study as it would have required a medium-term investment in human resources and a joint approach to be effective. Marie Stopes Ladies, an MSI global service delivery channel that has recently been started in Cambodia, is using such a strategy, by setting up a Facebook group to share information among the service providers.[8] In terms of interventions aimed at factory workers, we considered ways of direct provider-to-client communication, such as enrolling potential users of contraception through private providers or infirmaries, but opted against this due to concerns about maintaining communication, due to the high frequency of phone and/or SIM switching, or potential harm arising from other people listening to messaging, since phones were often accessible to several people aside from their owner.[27 28]

Given the widespread consumption of online videos, we considered that this medium could be the most effective in terms of reach and engagement, so we developed three short 'edutainment' videos about contraception. In addition, we adapted three informative videos made by MSI from English to the Khmer language, and adapted the MSI medical abortion 'Mariprist' instruction leaflet to a simple video format. These videos were not formally evaluated. The three short 'edutainment' videos were aimed at users or potential users of contraception aimed to address personal determinants, particularly knowledge and attitudes, towards contraception. Our objective was to frame contraception in a positive way, while acknowledging these frequently experienced and discussed context-specific side-effects such as body heat but being careful not to perpetuate them or dismiss them as imagined or unimportant. We felt that videos aimed at the general public that could be watched on mobile phones could be an effective way to deliver an intervention while limiting potential for harm. We decided to make three videos because female factory workers are not homogeneous. Working from the data we gathered from fieldwork, we constructed two 'personas', that is, fictional characters that represent characteristics associated with distinct groups of women.[29] The first persona represents married women, possibly with children, interested in spacing out births, and able to discuss contraception and abortion with their husbands. The second represents single women who may or may not be in a stable relationship, and who are interested in information about contraception and medical abortion, but do not necessarily discuss it with their partners.

**Table 2** Examples of possible interventions

| Example(s) | Intervention classification | Main target | Potential benefits | Potential challenges/ disadvantages |
|---|---|---|---|---|
| Edutainment video about contraception | Targeted client communication (transmit targeted health information to client based on health status or demographics) Goal is to increase awareness, not to instruct on specific details of contraceptive methods | Clients (potential or existing users of contraception, factory workers) | ► Potential for increased reach<br>► Potential to influence attitudes and behaviour about contraception use | Does not provide detailed information about pros and cons of contraceptive method |
| Provider-to-client communication | Targeted alerts/ reminders on mobile phones | Existing clients | ► Increase in follow-up visits and continuity of care<br>► Increase in adherence to instructions | ► Privacy (if phones are shared or anyway accessible to other people)<br>► Frequent changes of numbers and loss of phones make it difficult to have unique phone ID for unique clients |
| Provider-to-provider communication | Increase sharing of best practices in informal private online groups (eg, WhatsApp and Facebook group chats) | Providers, especially private providers who do not have regular opportunities for professional updates and training | ► Provide informal opportunities for sharing best practices and asking for advice<br>► Leverage social and informal ties to create strong 'communities of practice'[26] | ► Informal groups can be difficult to sustain without participants' buy-in; need a few motivated individuals<br>► Peer-to-peer information exchanges are not necessarily medically sound, so they could help spread misinformation |
| Instructional video about contraception methods | On-demand information services to clients | ► Existing clients<br>► Potential users already considering a specific method<br>► Providers who need reminding/training about how specific methods work | ► Video more attractive format compared with written text, and already a common source of information among targeted audiences<br>► Can reach clients who are not comfortable or able to go to pharmacies/clinics<br>► Comments on the videos can offer insights into frequent questions, and potentially serve as a source of referral for clinics | ► Unclear how likely videos are in influencing attitudes and behaviour<br>► Relies on being found in the midst of other commercial videos fighting for attention<br>► Relies on users having the connectivity to watch the video<br>► Requires resources to produce, post and keep updated<br>► Requires resources to potentially monitor and answer comments and questions |

Continued

**Table 2** Continued

| Example(s) | Intervention classification | Main target | Potential benefits | Potential challenges/ disadvantages |
|---|---|---|---|---|
| Video about abortion | Education about abortion | ▶ Clinic/pharmacy clients<br>▶ Women at risk of or with unintended pregnancy | ▶ Can reach audiences that are uncomfortable with text and/or with visiting clinics to ask for information<br>▶ Can offer a medically sound perspective, among propaganda and medically dubious videos currently available online | ▶ Difficult topic to engage with through an accurate, but accessible and engaging video<br>▶ Requires significant effort in managing the online presence of such videos (moderation, reliability, findability among competing anti-abortion videos, etc) |
| Instructional video about medical abortion | Education about medical abortion | Women with unintended pregnancy | ▶ More accessible alternative to written text for those with low-literacy levels<br>▶ Can be easier to access in private than written leaflets | ▶ Needs to be found online, against existing competing videos that might be less accurate but are ranked higher in search results<br>▶ Privacy issues, as it remains in search history |

We wrote terms of reference and invited local media agencies to pitch ideas. We then worked in collaboration with Phare agency to produce the videos.[15] This involved providing the agency with information on target audience and a visit to garment factory area for a pretesting workshop with garment factory workers to seek their feedback on some prototype videos and opinions on how to further improve the content. During this workshop, it was discussed that some of the factory workers would act in one of the videos. There was a two-way process by which the scripts were developed. English language versions of the scripts are shown in online supplemental file 2. The videos were filmed and Khmer subtitles added and the videos were released sequentially through MSIC's social media communication channels.[30 31] MSIC social media followers have grown significantly since 2019, so posting the videos on the MSIC Facebook and YouTube pages allowed us to reach a much broader audience than simply trying to distribute them among the factory workers we knew. The first video ('Mother') was aimed at married parous women, showing a conversation between a mother and her children whereby the woman was experiencing side-effects from using contraception with the children providing reassurance in a light-hearted way. The key message aimed to be that there is a contraceptive method that will fit with you. This was filmed by the agency in Battambang. The second video ('Love') was aimed at unmarried nulliparous women with the concept of a romance story being watched on a smartphone by a group of factory workers. The key message aimed to be being in

a romantic relationship and still having other life opportunities because of using contraception. The romance scenes were filmed in Battambang and the factory worker scenes were filmed in Phnom Penh. The third video ('Baby') was aimed at married or unmarried women with the concept of a Khmer karaoke-style comedy dance/song with a key message of finding the right method that fits with you and your family. We focused on these three topics as they emerged as the most important concerns for our diverse research participants in the qualitative research phase.

The videos were posted at 1 monthly intervals on Facebook and evaluated by Havas Champagne, a Cambodian media company after 1 month.[16] The videos were boosted on the MSIC Facebook page aimed to engage the target audience, female factory workers and reach the wider public. Geotargeted ads were used to identify and show content where most factories were located. A click to message campaign was also used to encourage people to send messages to MSIC Facebook page. The videos were placed on the news feed and Facebook suggested videos feed, and Facebook Watch feed. A shortened version rather than the original baby video was released. Overall, unique reach (number of people who saw a post a least once) was 3 462 176 with 2 839 255 engagements (number of people interacting with the content, for example, like, share, comment, reaction), 7 876 734 video plays, 16 000 000 impressions (number of people who saw a post, but may include multiple views by the same people), and 25 637 click to messages (number of people who click the

**Table 3** Key metrics of the three videos 1 month after release

|  | Mother | Love | Baby (shortened) |
|---|---|---|---|
| Video release | 13 Aug 2019 | 13 Sept 2019 | 11 Oct 2019 |
| Boost budget | $1016 | $1016 | $1166 |
| Video plays | 2 268 736 | 2 834 282 | 2 704 121 |
| Engagement and rate (eg, likes, shares, comments) | 679 591 | 1 265 398 | 873 388 |
| Engagement rate | 20% | 38% | 24% |
| Reach (nationwide) | 1 406 274 | 1 307 140 | 1 305 075 |
| Cost per reach (nationwide) | $0.30 | $0.32 | $0.32 |
| Click to Action (send message to Marie Stopes) | 3997 | 23 700 | 4728 |
| Cost per click | $0.03 | $0.01 | $0.03 |

Reach is the number of people who saw a post at least once. Engagement is the number of people interacting with the content. Example is like, share, comment, reactions. Click to Action measures the number of people who click the 'Send message' button that will lead to MSIC Facebook messenger.
MSIC, Marie Stopes International Cambodia.

'Send to Message' button that will lead to MSIC Facebook messenger). An automated response was set for those who sent a Facebook message and MSIC counsellors provided more comprehensive information regarding the topic they asked about and referred to MSIC clinics or services if required. Key metrics per each video are shown in table 3. In terms of click and engagement, the 'Love' video achieved the highest click to messages. Figure 3 shows trends in total calls to the MSIC contact centre and shows the number of Facebook messages sent to MSIC during the project period. Both figures show increases through September and October corresponding to the timing of release of the 'Love' and 'Baby' videos. Data were not collected from the MSIC contact centre or clinics regarding whether referrals were as a result of any specific content.

## DISCUSSION
In this paper, we have described the process by which we developed interventions to support reproductive

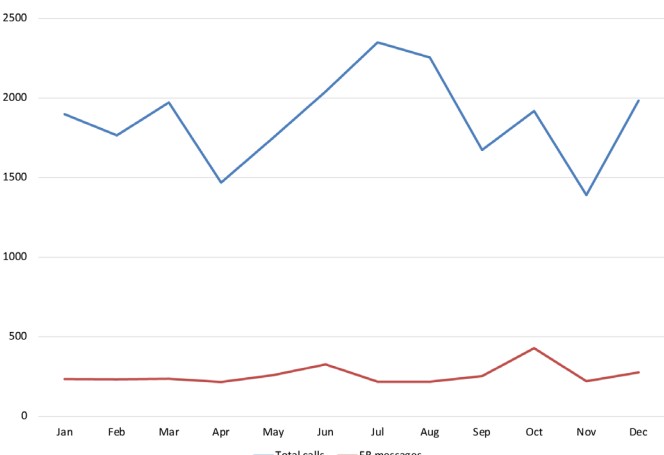

**Figure 3** Contact centre total calls and Facebook (FB) messages (inbound and outbound).

health of garment factory workers in Cambodia, primarily three short 'edutainment' videos about contraceptive methods. Our research identified some challenges that Cambodian garment factory workers experience regarding contraception and abortion. Our findings of unmet need for contraception due to social and structural constraints limiting women's opportunity to use contraception and their experience or concerns about side-effects are in keeping with previous literature.[22] Some side-effects reported, such as the experience of 'body heat' with the contraceptive pill, appear to be more common in Cambodia compared with elsewhere and possible reasons for this could be further explored. Future interventions should be sensitive to the finding that contraceptive side-effects may vary by context, and those critical to non-use may not be recognised in the WHO or other global literature and guidelines. Information about family planning often came from friends or family, and mass media campaigns in some cases. Use of private providers would in general result in a bias towards information about short-acting methods as few had the training to provide long-acting methods. Furthermore, providers may not have received formal family planning training and would reinforce myths or concerns about methods. Hence, we heard many reports of contraception use and subsequent discontinuation which could result in an unintended pregnancy. We considered that videos portraying contraception in a positive way while addressing context-specific issues could be an effective way to change attitudes and behaviour regarding contraception use.

Our three edutainment videos aimed to contain different key messages aimed at both married and unmarried women. By working with a media company, it was possible to incorporate fresh perspectives, and to involve factory workers in the production of the

videos. One of the main challenges was maintaining the key messages in the script when working in two languages. The videos were promoted through MSIC's communication channels and key metrics evaluated by an independent company. The videos had a greater reach and number of views if compared with other videos on the MSIC Facebook page, but it is difficult to know how much of this was due to the 'boosting' effect. The 'reach' of the videos was similar, however there were differences in the number of 'clicks' and 'engagement'. According to Havas/Champagne, this could be explained by the creative and message of the 'Love' video which was more relatable to younger audiences. MSIC saw an increase of inbound calls and Facebook messages especially in October, with messages doubled compared with previous month. The videos may have influenced people to directly reach out to MSIC contact centre for more information and led to subsequent use of services. However, data were not collected from the MSIC contact centre or clinics regarding whether referrals were as a result of the videos and therefore we cannot assume a causal relationship between our videos and the increased contact centre activity.

In general, it can be observed that the videos were less polished or professional in production if compared with other reproductive health videos such as produced by Population Services International Khmer or BBC Media Action.[6] [7] However, to our knowledge, they are unique in how they involved and addressed our targeted population in reflecting a narrative language that is familiar and appreciated and could serve as examples of videos that could be developed and evaluated further. Themes of love and romance in particular could be pursued further and we recommend collaboration with creative media companies to contribute fresh ideas. The feedback from Havas/Champagne media was to use videos with shorter length and less texts on the thumbnail could improve performance in terms of reach, watch time and higher chance of better audience retention. It is recommended to focus on mobile phones only, since there was virtually no traffic coming from other devices.

With regard to abortion, we observed that a variety of medical abortion products were widely available from the private providers, with variation in practice as observed elsewhere.[9] Withholding the leaflet when selling medical abortion was sometimes done to protect the client, according to the provider, but suggests 'information asymmetry' whereby the client is less informed and able to shop around. In any case, women purchasing medical abortion often had to rely on verbal information and remember how to take the product, what to expect and if they should return for follow-up. As we were not able to identify any reputable existing videos, in addition to the videos described in this article, we adapted a medical abortion leaflet

into a video format which can be viewed on the MSIC Facebook page.[30] This was not formally evaluated in the way the contraception videos were. The impact of providing support for medical abortion in different formats could be evaluated in a further study. In the meantime, our findings on leaflet retention and comprehension by pharmacy clients have contributed to an evidence brief published by MSI on supporting self-management of medication abortion from pharmacies.[32]

In this project, we did not develop any interventions directly aimed at providers. This was partly because it was difficult to establish the exact status of the providers in terms of current licence, approval to provide certain types of services and quality of medications. While we found varied adherence to best practice, it was encouraging that some providers expressed interest in further training or peer-support groups. Interventions to support or systems to inspect private providers should be explored further.

Strengths of this study include the interdisciplinary approach, collaborations between different academic institutions, an NGO and a media company, and the independent evaluation of the videos. This study has several limitations. Being a qualitative study, the number of participants was relatively small and is not representative of garment factory workers nor providers. Our goal was to reach a good variety of participants, but it is possible that the factories and providers who agreed to participate in the study had better conditions and services compared with other ones. Regarding those seeking abortion services, single women were under-represented in our interviews, and remain a very hard group to reach. Negative experiences of contraception may have been over-reported as many of the women we interviewed had experienced an unintended pregnancy. It is also possible that women's experiences of contraception, particularly side-effects, were difficult to translate from Khmer to English, although this was mitigated by having a Cambodian midwife and a linguist on the research team. Intervention mapping provides a rigorous detailed step-by-step approach to developing interventions.[33] In our experience, although we found elements of the approach useful, particularly in conceptualising the problem, other aspects were more challenging to apply. The range of local partners involved were limited to MSI, pharmacists, drug store owners and the factories, which combined with the limited time frame and funding for the project, constrained our options for the types of intervention we could consider. It was challenging to explain and translate details of intervention mapping across languages, cultures and disciplines working in different paradigms, especially as most of the team were unfamiliar with this approach. A strength of our work was the wide range of disciplines, cultures and perspectives we came from. As a result, our discussions identified similar or overlapping constructs which different people labelled in different ways. In the process of allowing the range of perspectives to contribute in ways that participants felt comfortable, we abandoned some of the intervention mapping steps which were experienced as too rigid,

formulaic and as coming from and using labels from one particular paradigm. We sought to employ participatory approaches in our needs assessment, so factory workers' views and local MSI staff were involved in developing the concepts for the videos. We obtained feedback from factory workers regarding the concept and content of videos allowing them to inform the process. Some factory workers volunteered to take part in a video. The process resulted in clearer documentation of interventions developed, but alternate interventions might have been developed using the approach applied more strictly or in a different way.

In conclusion, we describe the development of three 'edutainment' videos to address limitations in the provision of accessible reproductive health information and support use of family planning among garment factory workers in Cambodia. Key social media metrics of the videos were evaluated after 1 month. These videos could be further improved and additional videos could be developed. More work is required to develop appropriate and effective interventions to support reproductive health of garment factory workers in Cambodia.

**Author affiliations**

[1]School of Tropical Medicine and Global Health, Nagasaki University, Nagasaki, Japan

[2]Department of Clinical Research, London School of Hygiene and Tropical Medicine, London, UK

[3]Marie Stopes International Cambodia, Phnom Penh, Cambodia

[4]Marie Stopes International, London, UK

[5]Population Health, London School of Hygiene and Tropical Medicine, London, UK

[6]School of Languages, Cultures and Linguistics, School of Oriental and African Studies (SOAS), University of London, London, UK

[7]King's College London, London, UK

**Acknowledgements** We would like to appreciate many clients, factory managers and private providers who contributed to this study. We would also like to thank Marie Stopes International Cambodia for significant cooperation in the entire process of this study. Lastly, we would like to show our gratitude to Professor Tung Rathavy, Director of the National Centre for Maternal and Child Health Centre, for conceptualisation and inputs on this study.

**Contributors** CS contributed to the conception and design of the research, data acquisition, data analysis and interpretation, and drafted the manuscript. LS contributed to the conception and design of the research and data acquisition, and substantively revised the manuscript. CFET contributed to the conception and design of the research and substantively revised the manuscript. MM contributed to the conception and design of the research, data acquisition, data analysis and interpretation, and substantively revised the manuscript. CF contributed to the conception and design of the research and substantively revised the manuscript. JW contributed to the conception and design of the research. OA contributed to the acquisition of data, read and approved the final manuscript. CM contributed to data acquisition, analysis and interpretation. EO contributed to the conception and design of the research, data acquisition, data analysis and interpretation, and drafted the manuscript. The guarantor (CS) accepts full responsibility for the work and/or the conduct of the study, had access to the data, and controlled the decision to publish.

**Funding** This study was funded by the Arts & Humanities Research Council (AHRC) (number: AH/R006091/1) and a linked UK Research and Innovation Global Impact Accelerator Award (GIAA).

**Competing interests** None declared.

**Patient consent for publication** Not required.

**Ethics approval** Ethical approval was obtained from the Cambodia Human Research Ethics Committee (number: 094NECHR), the London School of Hygiene and Tropical Medicine (LSHTM) (number: 14646), and Marie Stopes International (MSI) Ethics Committee (number: 003-18). Informed written consent was obtained from all interview and focus group participants in this study.

**Provenance and peer review** Not commissioned; externally peer reviewed.

**Data availability statement** Data are available upon reasonable request. The data that support the findings of this study are available on request from the corresponding author [CS] by emailing christopher.smith@lshtm.ac.uk. The data are not publicly available because they contain information that could compromise research participants' privacy and consent.

**ORCID iDs**

Chris Smith http://orcid.org/0000-0001-9238-3202

Camille Florence Eder Tijamo http://orcid.org/0000-0002-8048-7941

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
