## [Reviewer comments · BMJ Open]

ARTICLE DETAILS

TITLE (PROVISIONAL)	Development of an intervention to support reproductive health of garment factory workers in Cambodia: a qualitative study
AUTHORS	Smith, Chris; Sokhey, Ly; Tijamo, Camille Florence; McLaren, Megan; Watkins, Justin; Amra, Ou; Masuda, Chisato; Oreglia, Elisa

VERSION 1 – REVIEW

REVIEWER	Tuot, Sovannary KHANA, Research Center
REVIEW RETURNED	12-Mar-2021

GENERAL COMMENTS	The paper is relevant to Cambodian context. However, it needs substantial revision. It includes concise the abstract, clear objective, clear methods and steps, clear and structure results and conclusion. Please see my comment direct to the pdf file. The reviewer provided a marked copy with additional comments. Please contact the publisher for full details.
--

REVIEWER	Mehta, Kala University of California, San Francisco
REVIEW RETURNED	University of California, San Francisco

GENERAL COMMENTS	This is a very elaborate process evaluation regarding short films for social media on the topic of contraception and abortion for women in Cambodia, primarily factory workers. This is an important topic and the mode of delivery through social media can increase access and reduce stigma, particularly in low and middle income countries. What impressed me about this article is the thoughtfulness in preparing and producing the videos and the background research. On balance, what perhaps needs strengthening is a more clear delineation of exactly what methods were used, in what sequence and a greater emphasis on the potential geographic transportability of such videos to other LMIC contexts. Page 3, lines 8-9 you could name this a process evaluation page 3, lines 40-44 were the videos only distributed on social media, why not through the factories themselves on breaks? page 3, lines 45-53, what was the acceptability impact and outcomes of the videos? page 4, lines 29-34 what reproductive and abortion strategies are cultural norms in Cambodia? How were these videos offering new or different strategies?
--

	page 6, lines 19-21 different researchser at different times? Maybe draw a timeline for an external audience? page 6, 30-32, what are digital cultures elaborate details on the methods. how many interviews? were meetings part of the study or were they to manage the study? page 6, lines 24-34 many methodologies used. the descriptions of these are intertwined. I suggest adding headers for each methodology used, its purpose in the larger framework and in the results reporting what peice of knowledge was gained from that methodology. This would go far towards potential reproducibility for example, did you actually conduct the ethnography first? then the needs assessment and then the intevention mapping? page 7, lines 7-10 "hanging out' colloquial page 7, 41-44, this is the main goal of the paper, should be in the title and abstract page 7, line 50 what were the precise barriers, define page 8, line 41 is not clear what is the relationship of the abortion seekers to the factory workers? page 10, lines 42-46 do you have an indication od the growth of smartphone/internet use in Cambodia? page 13, lines 3-13 what about engaging factory employers? could employees watch these and other health videos (maybe safety?) on breaks? page 13, line 26-31 did you consider other, less stigmatizing topics also on reproductive health like health changes during pregnancy or menopause? Discussion. I found the discussion good and helped clarify the importance of the findings. Woudl be fantastic to augment the potential use of these films or something akin in other LIMIC contexts? What comparisons can be made and how is this new and different from before? A formal outcomes analysis should ensue. If the authors address these topics, I think this manuscript would make a fine contribution to the field of reproductive health
--	--

VERSION 1 – AUTHOR RESPONSE

Mr. Sovannary Tuot, KHANA (Reviewer: 1) Comments to the Author:	
The paper is relevant to Cambodian context. However, it needs substantial revision. It includes concise the abstract, clear objective, clear methods and steps, clear and structure results and conclusion. Please see my comment direct to the pdf file.	Thank you for your comments. Whilst they are directly annotated on the PDF, in some cases it is difficult to know which part of the manuscript to attribute some of the comments. We have revised the manuscript to provide more clarity.
Study design was mixed up. It was not clear what approach they used. Possible formative participatory approach to develop the intervention (p4)	We have revised to provide more clarity but would also make the general point that this is a qualitative research paper incorporating different research methods rather than one main study design.

An overall, method looks mixed up and did not present clear. It needs to substantially rework as following: study design, setting, participants, method of data collection, data management and analysis. It will look great if we can develop a diagram on the intervention development process through all phases. (p7)	Thank you for this comment. We have re-written the Methods section (p5-7) to further clarify the methods, in what sequence they were used. We have included an additional figure (Figure 1) which indicates when each phase of the research was conducted.
You need to specific month. it reflects to the duration of data collection rather than the whole research project.(p7)	We have included the specific months in the Methods section (p5) and the additional figure (Figure 1) indicates when each phase of the research was conducted.
How many? (p7) [in reference to this statement: "a suburb of Phnom Penh where many garment factories cluster"]	We did not count the total number of factories in this area so cannot state a figure
What is this sentence link to research team?	We are not exactly sure what is meant by this comment, but we have added some additional information on the constitution and roles of our research team at the start of the Methods section
It could be method of data collection, including consultative meeting with experts. Also, it was not clear what method they used with whom? What general observation mean?	Observation is part of qualitative ethnographic research and described more fully in the text that follows on p6, the additional figure (Figure 1), and in more detail in the referenced study protocol (reference 11)
What design they actually used? it seems many over time: Need assessment, develop intervention modality, and validation of the intervention content. If it is so we can name as formative qualitative study which involved several steps as mentioned above. Please refine this.	Thank you. In the Methods (p5) we have described it as a qualitative study informed by intervention mapping which included a variety of research methods e.g. semi-structured interviews and participant observation etc.
Mapping interm of what? [in reference to "The end result was not a pure application of intervention mapping but rather a modified approach to it".]	Intervention Mapping is an established intervention development methodology that we introduce together with a reference in the preceding paragraph on p5: "The methods were informed by intervention mapping methods,(13) but in a more holistic manner, which took into consideration the different disciplines involved"
So confusing, again using another ethographic approach to narrating the lives of garment factory workers, what is the purpose of this? [In reference to: "In order to	Thank you for your comment. We believe that understanding holistically the lives of factory workers helps us understand more clearly the reason behind choices that might seem

learn about the wider context of factory workers' choices regarding reproductive health, we conducted a short ethnographic study of their lives, living conditions, work places, and digital technology use"]	illogical or unjustified. For example, we discovered that women might live in cramped rooms with other people, and thus might be reluctant to bring home any kind of pills that hints at an active sexual life, especially if they are not married. For example, a roommate finding the leaflet that contains instructions for MA can cause embarrassment or reputational damage, and therefore some women will not take the instructional leaflet when buying MA pills. These details can only be found through close observation and engagement with research participants.
National Ethics Committee for Health Research (NECHR).	Thank you. We have updated this in the Ethics statement.
Why it should be here? it could be part of the participant procedure. Need to state clear of study participant: private provider, garment factory worker...? [In reference to the Patient and public involvement statement]	A dedicated statement on Patient and public involvement is a requirement of the journal
It should be part of method. [in reference to: "We conducted 16 interviews with women seeking abortion from private providers located near the factories and 13 interviews with private providers working in different facilities such as infirmaries, pharmacies and bigger clinics. Findings from these interviews are reported elsewhere"]	Thank you. We have moved that text to the Methods section (p6)
Is it true? workers would have break time and choice of contraceptive method doesn't require to take during working hour. Or for example, injectable or monthly pill. [in reference to: "Factory workers' opportunities for using contraception to prevent pregnancy were constrained by their long working hours, limiting their ability to attend distant clinics with restricted opening hours e.g. for long acting contraception. Pharmacies close to their home"]	We have edited the paragraph in the Results section (p9) to clarify why nearby facilities weren't always a good choice: "Pharmacies close to their home and workplace were therefore the most practical option for healthcare seeking, but there was sensitivity and stigma related to obtaining contraception or MA near where women lived or worked. Women seeking supplies from drug stores or pharmacies would try to go to places far from where they lived, as to avoid being recognized, or would need to wait until no one else was at the counter or whisper to the provider to achieve any privacy. Young, unmarried women, sought privacy in both buying and taking in taking visible contraception such as the pill, because of was often essential due to negative social attitudes towards sexual activity for unmarried women, yet privacy was very difficult to achieve in cramped and shared

	living conditions, and in small neighborhoods where everybody knows everybody.”
All are garment factory workers or also include general women. Specific population might require different needs or facing with different barriers.	This is a good point. Our project was focused on garment factory workers and it is true that other populations might have different needs or face different barriers. It was beyond the scope of our project to assess this.
Need a quotes Lots of NGO working on RH current with garment factory workers include MSIC. You need to ensure the theme and sub-theme emerged separately along with quotes. two different context. we need to separate with quotes to support the findings It could construct as sub-theme of accessing information and training along with quotes.	Thank you for these comments in reference to the reporting of the interviews in the Results section. Findings from the interviews with women seeking abortion and private interviews are reported in detail with quotes in a separate paper, already published (reference 12). Another paper will explore the garment factory workers lives in more detail. There is not the space to add quotes in this paper as we are describing the summary findings from several research phases, but the focus is on the intervention development.
What is the main purpose of this section. Desk reiew on video about family planning, what the study aimed for? [in reference to: Videos about family planning in Cambodia]	The purpose of this section was to describe our review of available videos on family planning in Cambodia, as described in the Results (p10): “Given the observed widespread use of Facebook and Youtube among factory workers, and the preference for receiving information via images, video and voice rather than text that emerged from interviews and observations, we searched Youtube for information on contraception, abortion and medical abortion,

	using a variety of search terms that reflected both medical and more popular terms.”
It is quite hard to see the coherence, theme, subthem of the the findings along with quotes. It needs to returcure and rework entirely. Some paragraphs moved around. [in reference to the Intervention development section] Findings of what? Again, another section talking about Video while jumped to this section.	In this section on Intervention Development (p11-14) we are describing how we considered different interventions based on our research findings. In that respect it is different to previous sections based on interviews, observation etc and does not require themes, quotes etc. We feel that it is currently structured in a way that systematically goes through reasons why we didn’t opt for certain interventions and how we settled on the videos and how we approached making the videos. We have made some changes to this section for improved clarity.
“Supported citation?”	There are two requests for ‘Supported citation’ in the discussion section, but in both cases we feel that we are presenting our own ideas rather than statements that require a reference. “There seems to be some confusion that stems from the fact that we report the results of qualitative research (hence the “Findings” heading followed by what the reviewer feels are unsubstantiated claims, but are in fact the findings from our field work), and then move to the development of the videos as informed by those findings.
Can't see the concrete conclusion.	We have added some more detail to the Conclusion (p18) as follows: “In conclusion, in this paper we describe the development of three ‘edutainment’ videos to address limitations in the provision of accessible reproductive health information and support use of family planning among garment factory workers in Cambodia. Key social media metrics of the videos were evaluated after one month. These videos could be further improved and additional videos could be developed. More work is required to develop appropriate and effective interventions to support reproductive health of garment factory workers in Cambodia.”
Dr. Kala Mehta, University of California, San Francisco (Reviewer: 2) Comments to the Author:	

This is a very elaborate process evaluation regarding short films for social media on the topic of contraception and abortion for women in Cambodia, primarily factory workers. This is an important topic and the mode of delivery through social media can increase access and reduce stigma, particularly in low and middle income countries. What impressed me about this article is the thoughtfulness in preparing and producing the videos and the background research. On balance, what perhaps needs strengthening is a more clear delineation of exactly what methods were used, in what sequence and a greater emphasis on the potential geographic transportability of such videos to other LMIC contexts.	Thank you very much for this comment. We have re-written the Methods section (p5-7) to further clarify the methods, in what sequence they were used. In addition, we have included an additional figure (Figure 1) which indicates when each phase of the research was conducted.
Page 3, lines 8-9 you could name this a process evaluation	Thank you for this suggestion. Process evaluation is a good suggestion but we thought 'qualitative study' more reflective of the study that includes intervention development and process evaluation components, but all using qualitative methods.(Abstract, p2)
page 3, lines 40-44 were the videos only distributed on social media, why not through the factories themselves on breaks?	This is a good point, but because of the impressive growth of MSIC's social media, especially their Facebook page (which we detail in another article), we felt that we'd have a much greater reach by concentrating on social media. Gaining access to factories is not an easy feat, even when MSIC and our researchers have good connections, and the number of people we could have reached this way would not have justified the effort. Moreover, the only way to distribute videos in the factory would have been by mobile-to-mobile video transfer, which is cumbersome, and uses up precious storage space in our potential audience's phones. We edited the text in the Intervention Development section on p13 to summarize this as: "The videos were filmed and Khmer subtitles added and the videos were released sequentially through MSIC's social media communication channels. MSIC social media followers have grown significantly since 2019, so posting the videos on the MSIC Facebook and Youtube pages allowed us to reach a much broader audience than simply trying to distribute them among the factory workers we knew.."

page 3, lines 45-53, what was the acceptability impact and outcomes of the videos?	Thank you for this comment. Unfortunately, the independent evaluation of the videos was limited to key metrics and did not include acceptability. We have reported the available data.
page 4, lines 29-34 what reproductive and abortion strategies are cultural norms in Cambodia? How were these videos offering new or different strategies?	Thank you for this comment, We have added some text and supporting references to the Background (p4): “In Cambodia, reproductive health services are available through the public and private sectors, including non-governmental organisations (NGOs) and promoted through traditional and social media, and through community initiatives such as Marie Stopes Ladies.” We felt that our videos were different in the way they were developed and the focus on context specific barriers to contraception that arose from our formative research and hope that this is reflected in the Intervention Development section (p13) ‘acknowledging these frequently experienced and discussed context specific side-effects such as body heat but being careful not to perpetuate them or dismiss them as imagined or unimportant’
page 6, lines 19-21 different researcher at different times? Maybe draw a timeline for an external audience?	Thank you for raising this point - because of the number of people and approaches involved, and the constant iteration throughout the two years, we realize that the timeline of research can be a bit confusing to follow. We have added a figure (Figure 1) with a timeline that illustrates methods, fieldwork and iteration, which will hopefully clarify how the different phases unfolded.
page 6, 30-32, what are digital cultures elaborate details on the methods. how many interviews? were meetings part of the study or were they to manage the study?	We clarify what ‘digital cultures’ is in the Methods section on p5 (“that is, the study of how different people and groups engage with digital technologies”), and we have added the different meetings and their purpose in the timeline.
page 6, lines 24-34 many methodologies used. the descriptions of these are intertwined. I suggest adding headers for each methodology used, its purpose in the larger framework and in the results reporting what peice of knowledge was gained from that methodology. This would go far towards potential reproducibility for example, did you actually conduct the ethnography first? then the needs assessment and then the intevention mapping?	Thank you for this comment. We have revised the Methods section p5-7 (with tracked changes) so the sequence of research is more clear. In addition, the new Figure 1 indicates when the different phases of research were conducted. Hopefully this is more clear now.

page 7, lines 7-10 'hanging out' colloquial	“Hanging out” is a well-established ethnographic research method. We have clarified that in the text on p6: “After an initial period of what ethnographers call “hanging out,” that is unstructured time spent with research participants doing whatever they do, in order to become familiar with their lives and to give them time to get used to our presence, we conducted semi-structured interviews” (We have added a supporting citation: Salvador, Tony, Genevieve Bell, and Ken Anderson. "Design ethnography." Design Management Journal (Former Series) 10.4 (1999): 35-41.
page 7, 41-44, this is the main goal of the paper, should be in the title and abstract	Thank you for this comment. The Abstract text has been modified “In order to address limitations in the provision of accessible reproductive health information for factory workers, and given their observed widespread use of social media, we decided to make three ‘edutainment’ videos about family planning which were evaluated after one month.” The title reflects the title of the overall research project, although modified to include the study design.
page 7, line 50 what were the precise barriers, define	Thank you for this comment. Some of the key barriers (e.g. side-effects) are presented later in the Reproductive Health sub-section of the Results section on p9-10 and also in our other publication (reference 12) but do not repeat in detail to avoid duplication
page 8, line 41 is not clear what is the relationship of the abortion seekers to the factory workers?	Thank you for this comment. Hopefully the sentence in the Background section p4-5 sets the scene sufficiently: “This current study was conceived based on the prior observation that many independent private healthcare providers are located in the vicinity of garment factories, but not much is known about what kind of reproductive health services they provide, particularly medical abortion and contraception, and what is the experience of clients who visit them”
page 10, lines 42-46 do you have an indication of the growth of smartphone/internet use in Cambodia?	The data available don’t allow us to be specific in terms of smartphone, internet and social media use - they are counted in different and irreconcilable ways, e.g. the same statistic will say that half the population has internet access and 70% uses social media, which is clearly impossible. We have added on p11

	what we think is the most relevant information to this specific context, which is the number of social media users. “the increasing number of mobile internet and social media users in the country, estimated to have reached in January 2021 12 million over a population of 16.83 million” (We have added a supporting citation: https://datareportal.com/reports/digital-2021-cambodia We Are Social, Hootsuite, 2021b. Digital 2021 Cambodia (January 2021) v01.)
page 13, lines 3-13 what about engaging factory employers? could employees watch these and other health videos (maybe safety?) on breaks?	We considered ways of getting factories involved, but concluded that it would put too much of a burden on both the employer and the employees, and potentially backfire in terms of the willingness of workers to watch the videos. Break times are limited, and they are the only times when workers can socialize with each other, or check their phones (which can't always be used on the factory floor). Making the videos a 'chore' imposed by the employer would be counterproductive (and possibly illegal).
page 13, line 26-31 did you consider other, less stigmatizing topics also on reproductive health like health changes during pregnancy or menopause?	We clarified how we selected these topics on p14: “We focused on these three topics as they emerged as the most important concerns for our diverse research participants in the qualitative research phase.” See also the paragraph on p13 on personas for why we chose these topics - they were a good fit with the personas we created from field data. “Working from the data we gathered from fieldwork, we constructed two “personas,” that is fictional characters that represent characteristics associated with distinct groups of women.(25) The first persona represents married women, possibly with children, interested in spacing out births, and able to discuss contraception and abortion with their husbands. The second represents single women who may or may not be in a stable relationship, and who are interested in information about contraception and medical abortion, but don't necessarily discuss it with their partners.”
Discussion. I found the discussion good and helped clarify the importance of the findings. Would be fantastic to augment the potential use	Thank you for the enthusiastic support! We are writing a separate paper that deals exclusively with health providers' social media strategies

of these films or something akin in other LIMIC contexts? What comparisons can be made and how is this new and different from before? A formal outcomes analysis should ensue.	and health-seeking behaviors, as there is a lot of activity in this area. In the course of the research for this other paper, we looked at the different strategies of NGOs in different LMICs, and discovered that there are so many different challenges and concerns that it is difficult at this point to generalize on lessons learnt or specific strategies. It'd be definitively an interesting area of future research!
If the authors address these topics, I think this manuscript would make a fine contribution to the field of reproductive health	Thank you very much for this comment